# The Prevalence and Correlated Factors of Occupational Stress, Cumulative Fatigue, and Musculoskeletal Disorders among Information Technology Workers: A Cross-Sectional Study in Chongqing, China

**DOI:** 10.3390/healthcare11162322

**Published:** 2023-08-17

**Authors:** Bailiang Zheng, Fengqiong Chen, Jin Wang, Huaxin Deng, Jinshan Li, Chunmin Zhou, Mengliang Ye

**Affiliations:** 1School of Public Health, Chongqing Medical University, Chongqing 401331, China; bailiang_zheng@stu.cqmu.edu.cn (B.Z.);; 2Institute of Occupational Health and Radiation Health, Chongqing Municipal Center for Disease Control and Prevention, Chongqing 400042, China; cqcdczwschenfq@163.com (F.C.);; 3National Institute for Occupational Health and Poison Control, Chinese Center for Disease Control and Prevention, Beijing 100050, China; wangjin@niohp.chinacdc.cn

**Keywords:** occupational stress, fatigue, musculoskeletal diseases, information technology, cross-sectional studies

## Abstract

Occupational stress, cumulative fatigue, and work-related musculoskeletal disorders (WMSDs) are major concerns in the field of occupational health. Previous studies on occupational health focused on key industries, such as medical care, while there were few related studies on Information Technology (IT) industry. Our study explored the factors influencing occupational stress, cumulative fatigue, and musculoskeletal disorders in the IT industry. We collected 1363 IT workers’ valid questionnaires, of which 73.1% of participants were technicians in Chongqing, 2021. The core occupational stress scale (COSS), self-diagnosis checklist for the assessment of workers’ accumulated fatigue and Chinese musculoskeletal disorders questionnaire were used to measure the occupational stress, cumulative fatigue, and musculoskeletal disorders of the respondents. Logistic regressions were used to explore the correlated factors affecting these indicators. The results showed that the prevalence of occupational stress, cumulative fatigue, and musculoskeletal disorders was 50.4%, 47%, and 72.3%, respectively. Working in the current position for 3–10 years was a common increased risk for all three indicators. Insomnia was associated with an increased risk of cumulative fatigue (Odds Ratio, OR = 2.156, *p* < 0.001) and musculoskeletal disorders (OR = 1.878, *p* < 0.001). Cumulative fatigue was also associated with an increased risk of having WMSDs (OR = 3.207, *p* < 0.001). According to our findings, occupational factors, such as long working years, overtime work, and personal lifestyle, are highly related to the occurrence of occupational stress, cumulative fatigue, and musculoskeletal disorders for IT industry workers. More attention should be paid to women and those with long working hours in the IT industry.

## 1. Introduction

Occupational stress and cumulative fatigue are widely recognized as important hazards to the workplace population and issues that require the attention of health regulatory authorities [1]. Excessive occupational stress may cause physical and psychological damage to individuals, such as anxiety or burnout. It also negatively affects worker’s productivity, job, and life satisfaction [2,3]. Gu et al. point out that occupational stress caused by high-intensity work, irregular work–rest schedule, and exposure to harmful factors, such as high temperature and noise, may be highly related to the occurrence of hypertension in petrochemical workers [4]. In some special industries and professions, such as the medical staff, around 75% of newly recruited nurses leave the job due to high work pressure [5]. During the COVID-19 pandemic, U.S. Mariners experienced higher stress and mental health problems than usual due to long working hours, distance from home, communication barriers, and other reasons [6].

It has been studied that stress had associations with burnout and fatigue and would have a long-term negative impact on an individual’s psychological health [7]. Regarding Work-related fatigue, it could accumulate and would cause many physical and mental problems, especially for workers in specific industries. A study on cumulative fatigue and mental disorders in bus drivers finds that 35.5% of the 965 surveyed drivers have high cumulative fatigue scores without previous mental disorders. For those who had previously experienced mental illness, anxiety, or mood disorders, the rate increases to 43.5% [8]. This study reveals the association between psychological problems and cumulative fatigue. Another study examines the association between mental health, sleep, and fatigue in 3144 professional firefighters from the eastern coast area of China. This study investigates physical and mental occupational fatigue in firefighter populations. The results show that fatigue is highly related to sleep quality, as well as mental health [9]. Physical and mental fatigue are positively associated with mental health problems.

Severe working stress would also have a significant correlation to worker’s different body parts and their musculoskeletal disorders [10], which is another huge risk to their health. Work-related musculoskeletal disorders (WMSDs) generally refer to a series of diseases caused by occupational activities characterized by repetitive movements, long working hours, or forced posture. Its prevalence is highly and largely related to ergonomic factors found in the workplace [11]. Work-related musculoskeletal disorders are commonly found in certain occupational groups. For example, a review of factors associated with musculoskeletal disorders among regular and special education teachers indicates that special education teachers had higher rates of musculoskeletal disorders than regular teachers because of the potential need to assist students in moving, feeding, changing diapers, and helping them walk. At the same time, age, working years, and fixed movements, such as frequently raising arms, increase the risk of musculoskeletal disorders for teachers [12]. Similar problems arise in the office sedentary population. A study on the risk factors for reducing musculoskeletal disorders among 679 office workers in the petrochemical industry in Thailand aged 20–59 years shows that people who were older, having a high Body Mass Index (BMI), seated for more than 4 h per day, and do not change their standing or sitting posture frequently have a higher risk of developing musculoskeletal disorders than others. Accordingly, changing these habits can reduce the risk of developing musculoskeletal disorders [13]. Additionally, implementing proper measures and solutions from an ergonomic aspect would relieve musculoskeletal disorders, a literature review pointed [14].

China has always been overseeing occupational health issues. The plan of “Healthy China Action (2019–2030)” also states the goal of occupational health, emphasizing the “control of excessive fatigue and work-related musculoskeletal diseases” [15]. As an emerging industry in the past two decades since the beginning of the 21st century, the IT industry attracts job seekers to the industry due to its relatively high income, more open and inclusive working atmosphere. However, the occupational health problems of IT workers, accompanied by those typical characteristics like keeping one posture for a long time, have gradually become prominent. The purpose of this paper is to understand the work and life characteristics of employees in various typical jobs in the IT industry, explore the related factors affecting occupational stress, cumulative fatigue, and musculoskeletal health of IT workers, and explore the influence of occupational stress and work-related fatigue on muscle health, in order to give more targeted suggestions for the occupational health of IT workers and promote the better designing of occupational health plans for the IT industry.

## 2. Materials and Methods

### 2.1. Research Design and Participants

This was an investigation of occupational hazards for key populations led by the Institute of occupational health and poison control, Chinese center for disease control and prevention. This was a cross-sectional study conducted in Chongqing in 2021 as a part of the national investigation. Chongqing has been striving to build a data center cluster in Southwest China due to the emergence of many internet industry companies. The following formula for calculating the sample size was used: $n=\frac{{\left(Z_{\alpha /2}\right)}^{2}\pi (1-\pi )}{\delta^{2}}$, where *n* was the sample size, $Z_{\alpha /2}$ was the standard normal distribution at 1 − *α*% confidence level (*α* = 0.05, 95% = 1.96), *π* was the estimated proportion of previous evidence-based practice (22.1% in our case), δ was the margin of sampling error tolerated (=0.1**π*). Including 10% non-response rate, the sample size should be at least 1500. The study comprised 1510 Information Technology workers from 16 internet service enterprises. 

The enterprises were randomly selected by the enterprise scale, and the participants were recruited via cluster sampling. They worked in 4 major types of work: as technicians (e.g., programming, data processing, software development, etc.), salespersons, in customer service, and supportive administration (e.g., secretary, accountant, etc.). They were recruited voluntarily, had worked in their current position for more than half a year and did not have a history of psychiatric disorders or of taking antipsychotic drugs. The exclusion criteria were as follows: working for less than half a year, taking sick leave and not returning to work for more than 1 month, taking antipsychotic drugs. Answering sheets with more than 10% missing values were excluded from analytic process. In the end, a total of 1363 valid questionnaires were collected. Overall, 73.1% of total participants were technicians. This investigation was previously reviewed by the medical ethical review committee of the national Institute for occupational health and poison control, Chinese center for disease control and prevention.

### 2.2. Demographic, Working and Living Characteristics

Sociodemographic characteristics included date of birth, sex, education level, marital status, monthly income, job type, total working years, and current working years. Monthly income covers less than 465.63 USD a month, 465.63 to 775.90 USD a month, 776.05–1086.32 USD a month, 1086.47–1396.74 USD a month, 1396.89–1707.16 USD a month, equal or more than 1707.31 USD a month. Job types mainly included technician, salesperson, customer support staff, and administrative staff. The job characteristics survey included the “average daily working hours in the past month”, “average daily working days in the past month”, “average daily overtime hours”, “whether the work requires shift work”, and “whether the work requires night shift”. All the detailed response options can be seen in the second table, item column.

Living behaviors and lifestyle included “have you ever smoked”, frequency of smoking, alcohol consumption, frequency of drinking alcohol, frequency of outdoor activities for more than 30 min, frequency of at least 30 min of high-intensity exercise, and frequency of at least 30 min of medium-intensity exercise. At the same time, the life behavior and lifestyle survey were conducted to determine whether the respondents had sleep problems. 

### 2.3. Assessment of Occupational Stress

The Core Occupational Stress Scale (COSS) developed by the Chinese Center for Disease Control and Prevention was used to evaluate the occupational stress of the respondents. It has been proven valid and reliable in Chinese populations [16]. The scale has been tested for reliability and validity and could be used as an occupational stress assessment tool for key occupational populations in China. The scale consisted of four dimensions of “social support”, “organization and reward”, “demand and effort”, and “autonomy” with a total of 17 items. Each item was scored using a 5-point Likert scale with 1 for “strongly disagree”, 2 for “disagree”, 3 for “mostly agree”, 4 for “agree”, and 5 for “strongly agree”. First, two dimensions were transformed by “6 minus actual score”, the sum score of the scale and the scores of each dimension were calculated, respectively. The higher the total score was, the more serious occupational stress was. Based on previous studies, a total score greater than or equal to 50 was associated with occupational stress.

### 2.4. Assessment of Cumulative Fatigue

The Self-diagnosis Checklist for Assessment of Workers Accumulated Fatigue was used to measure the degree of fatigue accumulation, which was developed by the Ministry of Health, Labor, and Welfare of Japan. It has been proven to have a good reliability and validity [17,18]. The questionnaire consists of two major parts. The first part is about “the evaluation of self-assessed symptoms in the last month”, which is divided into four grades according to the score ranging from i to iv, that 0~4 is level I, 5~10 is level II, 11~20 is level III, equal or more than 21 is level IV. The second part is “evaluation of work performance in the last month”, which is composed of seven items, and the total score of each item is also divided into four grades: 0 is grade A, 1 to 2 is grade B, 3 to 5 is grade C, and more than 5 is grade D. Combined with the above two parts of the rating, the “workload scoring criteria” was scored, and the specific criteria are shown in Table 1. Score 0~1 means low workload, 2~3 means slightly high workload, 4~5 means high workload, and 6~7 represents extremely high workload. It was believed that if the workload score is between 2 and 7 points, there is a manifestation of fatigue accumulation.

### 2.5. Assessment of Work-Related Musculoskeletal Disorders (WMSDs)

For work-related musculoskeletal disorders, Chinese Musculoskeletal Disorders Questionnaire was applied to investigate the occurrence of WMSDs. It was previously developed from Nordic Musculoskeletal Questionnaire. The questionnaire has been tested and has good reliability and validity [19,20]. A questionnaire was used to investigate the symptoms of injuries in nine body parts in the past year, including neck, shoulder, back, elbow, waist, wrist, hip, knee, and ankle. The severity of WMSDs and absence due to illness were evaluated. It was generally considered to have Musculoskeletal disorders if symptoms of the disorder occur at one or more parts in this survey. 

### 2.6. Statistical Analysis

Descriptive statistics was used to summarize the basic characteristics of the respondents. The score was described by mean with standard deviation. For the quantitative data that met the normal assumption, independent sample t-test and one-way analysis of variance were used for univariate statistical inference. The qualitative data that did not meet an assumption of normal distribution were tested by Mann–Whitney U and Kruskal–Wallis H method [21]. As for the correlation among occupational stress, cumulative fatigue and work-related musculoskeletal disorders, Spearman correlation coefficient was used to measure the correlation among the original scores of these three indicators.

Binary logistic and ordinal logistic regression were used for multivariate analysis. After classifying the participants with occupational stress, cumulative fatigue, and WMSDs according to the definition, the variables with differences in the above univariate analysis were included in the model to explore the influence of each indicator with changes of confounding factors. Occupational stress and cumulative fatigue were also included in the model to assess the impact on WMSDs. Male, excessive drinking every day, aged between 20 and 25 years old, single, monthly income less than 465.63 USD, total working time of 0–5 years, current working time of 0–1 years, never walking or riding outside, no high-intensive exercise, no medium-intensity exercise, working less than 40 h per week, no overtime every day, no insomnia symptoms were control characteristics, respectively.

IBM SPSS Statistics 26.0 (IBM, Armonk, NY, USA) was used to conduct the statistical analyses, with a two-tailed probability value of <0.05 considered to be statistically significant.

## 3. Results

### 3.1. Characteristics and Univariate Analysis for Influencing Factors of Occupational Stress, Cumulative Fatigue, and Work-Related Musculoskeletal Disorders

To explore the influence of personal characteristics and living behaviors on occupational stress, cumulative fatigue, and work-related musculoskeletal disorders, Table 2 reflected the basic characteristics of respondents, the average scores of three indicators and the difference (standard deviation) of these factors. Univariate analysis was adapted for the influence of the above demographic characteristics and living behaviors on occupational stress, cumulative fatigue, and musculoskeletal disorders.

Concerning occupational stress, there were differences in gender, night shift, monthly income, current working years, weekly working hours, and overtime work hours. Those not working night shifts, those who had a current job for less than one year, and those who did not work overtime had higher occupational stress than other participants in the same group (*p* < 0.05). Sex, monthly income, weekly work hours, daily overtime work were significantly more associated with occupational stress (*p* < 0.001).

In terms of cumulative fatigue, workload scores over two were considered to indicate the presence of cumulative fatigue. The higher the score, the more severe cumulative fatigue the worker had. The results showed that drinking alcohol, the need to work night shifts, an education level of middle school or below and a master’s degree or above, married but separated, having a monthly income of more than 1396.89 USD, working for more than 15 years, being at the current work for 3 to 10 years, smoking over 10 cigarettes a day, drinking three to four times a week or drinking every day, never engaging in any outdoor exercise, never doing any medium-intensive exercise had significantly severe cumulative fatigue (*p* < 0.05). However, weekly working hours, daily overtime work, and insomnia were significantly more associated with cumulative fatigue (*p* < 0.001).

Regarding work-related musculoskeletal disorders, the results showed that most of the respondents had more than one musculoskeletal disorder. The results indicated that female, drinking alcohol, aged between 30 to 35 years old, married and living together, monthly income of more than 1396.89 USD, total work of 5 to 10 years, current work life of 3 to 10 years, never performing any outdoor exercise, medium- and high- intensity exercise all had serious musculoskeletal disorders (*p* < 0.05). Similarly to cumulative fatigue, weekly working hours, daily overtime work, and insomnia were significantly more associated with musculoskeletal disorders (*p* < 0.001).

### 3.2. Correlation Analysis of Occupational Stress, Cumulative Fatigue, and Work-Related Musculoskeletal Disorders

In Table 3, the results of correlation analysis showed that the correlation coefficient between each two values of occupational stress, cumulative fatigue and work-related musculoskeletal disorders was less than 0.5, which means the correlation relationship was not strong. There was a negative correlation between occupational stress and cumulative fatigue and work-related musculoskeletal disorders. Meanwhile, a positive correlation between cumulative fatigue and musculoskeletal disorders existed.

### 3.3. Multivariate Analysis for Influencing Factors of Occupational Stress, Cumulative Fatigue, and Work-Related Musculoskeletal Disorders

Based on single factor analysis, a binary logistic regression model was used for occupational stress and cumulative fatigue analysis, while a ordinal logistic regression model was used for musculoskeletal disorders to further explore the changes in risk factors with these confounding factors, considering those three scores were used as dependent variables. Musculoskeletal disorders were grouped as no musculoskeletal disorders (score 0), mild musculoskeletal disorders (score 1 to 2), and severe musculoskeletal disorders (score 3 or above).

Table 4 shows the risk factors for occupational stress, cumulative fatigue, and work-related musculoskeletal disorders. People who had worked in their current position for 1 to 3 years (OR = 1.567, 95%CI: 1.158–2.122) and 3 to 10 years (OR = 1.599, 1.173–2.179) had a greater risk of occupational stress than those who had their current job for less than 1 year. Workers who worked overtime every day had a higher risk of occupational stress than those who did not. The Odds Ratio of working overtime for 1–2 h a day, 3–4 h a day, and more than 5 h a day were 2.113 with 95% confidence interval from 1.555 to 2.870, 3.560 with 95% confidence interval from 2.24 to 5.658 and 2.629 with 95% confidence interval from 1.539 to 4.492. The results showed that the risk of occupational stress was highest when having 3–4 h overtime work a day.

As for cumulative fatigue, working in the current position for 3 to 10 years was associated with a higher risk of cumulative fatigue than those who had worked in their current position for less than 1 year. Medium-intensity exercise 1 to 3 times a month (OR = 0.727, 95%CI: 0.542–0.975) and 1 to 3 times a week (OR = 0.545, 95%CI: 0.342–0.868) were protective factors against cumulative fatigue. People who worked 50 to 59 h a week (OR = 2.221, 95%CI: 1.152–4.282) or more than 60 h a week (OR = 3.668, 95%CI: 1.871–7.191) had a higher risk of cumulative fatigue than those who worked less than 40 h a week. People who had to work overtime, no matter how long it was, were more likely to suffer from cumulative fatigue than those who did not work overtime. The risk increased with the increase in everyday overtime hours (OR = 2.155, 4.289, 21.075, respectively). Participants with insomnia had a higher risk of cumulative fatigue than those without insomnia (OR = 2.156, 95%CI: 1.659–2.801).

Considering influencing factors of work-related musculoskeletal disorders independently, variables with *p* values < 0.05 in the bivariate analysis were subsequently included in an ordinal logistic regression analysis. Women had a 2.305 times risk of WMSDs than men (OR = 2.305, 95%CI: 1.780–2.986). The risk of having WMSDs when having worked more than 15 years is 2.032 times higher than those with 0 to 5 employment years (95%CI: 1.060–3.896). People working their current job for 1 to 3 years and 3 to 10 years had a 1.344 (95%CI: 1.020–1.772) times and 1.366 (95%CI: 1.024–1.821) times higher risk of WMSDs than those engaged in their current job for less than 1 year. Moderate high-intensity exercise was a protective factor against WMSDs, as the study showed that people who exercised 1 to 3 times a month had a 0.680 times lower risk of WMSDs than those who never performed high-intensity exercise (OR = 0.680, 95%CI: 0.511–0.903). Working overtime is an obvious risk factor. The risk of WMSDs for 1 to 2 h overtime work per day, 3 to 4 h per day, and more than 5 h per day were 1.741 (95%CI: 1.322–2.294), 2.049 (95%CI = 1.346–3.119), and 1.913 (95%CI:1.168–3.134) times higher than those who did not work overtime, respectively. People with insomnia had a higher risk of WMSDs than those without insomnia symptoms (OR = 1.878, 95%CI: 1.507–2.340). In Table 5, after adding occupational stress and cumulative fatigue into the model, the risk of having WMSDs for females is still 2.435 times higher than for males (95%CI: 1.872–3.166). Drinking alcohol and 1–2 h overtime work are also associated with an increased risk of having WMSDs. Occasional 1–3 times high-intensity exercise a month had a 0.639 times risk of having WMSDs compared to not doing any high-intensity exercise (95%CI: 0.478–0.855). As for occupational stress, it was not significantly associated with WMSDs. However, having cumulative fatigue increased the risk of having WMSDs significantly(*p* < 0.001). Those who had cumulative fatigue had a 3.207 times higher risk of having WMSDs than those who had no cumulative fatigue symptoms (95%CI: 2.511–4.096).

## 4. Discussion

Our study found that 50.4% of the respondents had occupational stress symptoms. This indicator is significantly higher than in other industries [22,23,24], which means occupational stress of the IT worker needs more social attention. The study found that current working years from 1 to 10 and an increase in overtime working hours per day were associated with an increased risk of experiencing occupational stress, which was in line with the study of Siu and Sun et al. [25,26]. Working over 60 h per week also causes a higher risk of experiencing occupational stress compared to those who work less than 40 h a week. Additionally, lower support from supervisors and co-workers, lower decision latitude with respect to work, and higher psychological job demands could lead to occupational stress and related health issues. Necessary protection and occupational disease management for populations with similar occupational characteristics could effectively improve their happiness [27]. 

Concerning cumulative fatigue, 47% of the respondents had symptoms of cumulative fatigue, and 22.7% of the total participants had serious symptoms of cumulative fatigue. This index was also higher than the average level of other industries [28,29,30]. Stress and fatigue might have some correlation. In a previous literature review, 226 articles were included to examine the effects of shift work on sleep among paramedics, which stressed that working in a highly stressful environment could increase work-related fatigue and potentially cause a disaster [31]. Like occupational stress, 3 to 10 current working years and over 60 h working a week were associated with a higher risk for cumulative fatigue. Another big factor that led to cumulative fatigue was daily overtime work hours. The longer the hours, the higher risk of cumulative fatigue. Workers performing over 5 h of overtime per day had, on average, a 21.075 times higher risk of experiencing cumulative fatigue. Those with symptoms of insomnia also have a higher risk of experiencing cumulative fatigue than those that do not. Severe fatigue might have persistent physical and psychological damage to workers [32]. At the same time, it also reduces workers’ working efficiency [33] and increases the possibility of safety accidents [34]. It was a welcome finding that a certain amount of medium-intensity exercise is a protective factor against accumulated fatigue. For sedentary people with long working hours, moderate exercise could effectively reduce the risk of cumulative fatigue [35,36]. 

In terms of work-related musculoskeletal disorders, 72.3% of the respondents reported that they had musculoskeletal disorders symptoms, and 45% of the total respondents had severe musculoskeletal disorder symptoms. Those with cumulative fatigue symptoms have, on average, a 3.207 times higher risk of developing WMSDs, which showed that stress, musculoskeletal disorders, and fatigue may have some potential associations. In a previous study, work–family conflict led to stress, then caused musculoskeletal disorders, and finally, led to fatigue [37]. In our study, we found that women had a higher risk of getting musculoskeletal disorders than men, which might be related to the lower physical adjustment and physical recovery ability of women [38,39]. Occasional high-intensity physical exercise could relieve the pressure of experiencing WMSDs, according to the results. Based on previous studies, exercise had already been considered to effectively reduce fatigue [40] and had been applied to the rehabilitation treatment of rheumatoid arthritis [41], which had certain reference meaning for IT workers’ prevention of WMSDs. 

### Limitations

This study is a cross-sectional study, and causality between indicators cannot be inferred. Although there is no obvious linear correlation between the three indicators, there may still be other relationships. In future studies, a more complex and detailed relationship between occupational stress, work-related fatigue, and musculoskeletal disorders can be further explored. In addition, although the research has investigated obvious occupational environment factors and characteristics of the IT industry, there is a lack of comprehensive summary of other factors. For instance, living habits like living with family or colleagues or alone, and frequency of communication with neighbors would have some associations with loneliness and possibly impact occupational stress and fatigue level. Additionally, special circumstances like exposure to COVID-19 may be associated with a higher risk of having fatigue, stress, or other psychological well-being issues, with which we do not make comparison assessments in this study. In terms of sample selection, the exclusion of 147 participants for the analysis process may cause bias in the results. Meanwhile, the survey focuses on the limited IT enterprises in Chongqing, which lacks a certain representability in the reflection of the whole IT industry population.

## 5. Conclusions

In summary, IT workers have a high prevalence of occupational stress, cumulative fatigue, and musculoskeletal disorders symptoms. We find that working in the same position for a relatively long time and long weekly working hours are significant risk factors for these three symptoms. Another work-related common risk factor is daily overtime work. In cumulative fatigue, the longer the everyday overtime work, the higher the risk. As for occupational stress and WMSDs, 3 to 4 h of overtime work per day leads to a higher risk of experiencing these two symptoms. Luckily, occasional medium- and high-intensity exercising are significantly adverse to getting cumulative fatigue and WMSDs. However, around 29% of the participants reported that they would perform such physical activities one to three times a month. Over half of the participants said that they did not perform any medium- or high-intensity exercise. As a result, employers and social security departments need to focus on characteristics such as long working hours and frequent overtime work, formulating corresponding countermeasures from policy making to individual behaviors to reduce the physical and psychological burden of IT industry employees and improve their occupational health status.

## Figures and Tables

**Table 1 healthcare-11-02322-t001:** Workload Scoring Criteria.

		Work Performance			
		A	B	C	D
**Self-Assessed Symptoms**	**I**	0	0	2	4
	**II**	0	1	3	5
	**III**	0	2	4	6
	**IV**	1	3	5	7

**Table 2 healthcare-11-02322-t002:** Characteristics and scores of occupational stress, cumulative fatigue, and WMSDs among participants.

				Occupational Stress			Cumulative Fatigue			WMSDs		
Item	Group	n	%	Total/Score	SD	*p*	Total/Score	SD	*p*	Total/Score	SD	*p*
Sex	Male	993	72.9	51.33	0.18	**	1.88	0.06	**	2.46	0.08	0.001 *
	Female	370	27.1	52.69	0.28		1.45	0.09		2.99	0.13	
Rotate shift	No	1340		51.71	0.15	0.684	1.76	0.05	0.787	2.62	0.07	0.109
	Yes	23		51.22	1.64		1.65	0.31		1.78	0.42	
Drinking Alcohol	Yes	1011		51.67	0.17	0.791	1.86	0.06	0.001 *	2.69	0.08	0.046 *
	No	352		51.77	0.33		1.47	0.10		2.38	0.13	
night shift	No	1222		51.86	0.16	0.010 *	1.67	0.05	**	2.58	0.07	0.245
	Yes	141		50.32	0.57		2.53	0.19		2.84	0.23	
Age	20–25	20–25	18.0	52.18	0.37	0.314	1.36	0.11	**	2.26	0.16	0.035 *
	>25–30	>25–30	36.1	51.75	0.26		1.73	0.09		2.60	0.11	
	>30–35	>30–35	23.8	51.66	0.34		1.82	0.11		2.88	0.14	
	>35	>35	22.2	51.26	0.29		2.07	0.12		2.60	0.14	
Education Level	Middle School or Below	5	0.4	48.60	2.40	0.662	2.00	0.63	0.004 *	3.80	1.56	0.057
	High School	19	1.4	51.16	1.42		1.68	0.45		2.00	0.50	
	Associate or Diploma	497	36.5	51.58	0.26		1.53	0.08		2.44	0.11	
	Bachelor	779	57.2	51.77	0.20		1.85	0.07		2.67	0.09	
	Master or above	63	4.6	52.16	0.67		2.38	0.26		3.24	0.31	
Marital Status	Single	663	48.6	52.05	0.21	0.102	1.58	0.07	0.002 *	2.40	0.10	0.017 *
	Married and living together	614	45.0	51.41	0.23		1.90	0.08		2.83	0.10	
	Married but separated	63	4.6	50.68	0.81		2.30	0.24		2.71	0.31	
	Divorced or widowed	23	1.7	51.91	1.06		1.78	0.39		2.26	0.46	
Monthly income	<465.63 USD	27	2.0	53.93	0.95	**	1.19	0.28	**	2.37	0.50	0.002 *
	465.63–775.90 USD	228	16.7	53.16	0.35		1.01	0.10		2.08	0.16	
	776.05–1086.32 USD	316	23.2	51.60	0.33		1.81	0.11		2.87	0.14	
	1086.47–1396.74 USD	256	18.8	50.95	0.36		1.90	0.12		2.53	0.16	
	1396.89–1707.16 USD	171	12.5	50.50	0.45		2.05	0.15		3.01	0.19	
	≥1707.31 USD	365	26.8	51.79	0.29		1.99	0.10		2.59	0.12	
Work as	technician	997	73.1	51.86	0.18	0.116	1.78	0.06	0.464	2.55	0.08	0.656
	Salesperson	59	4.3	49.83	0.78		2.00	0.27		2.64	0.30	
	Customer support staff	49	3.6	51.39	1.03		1.41	0.24		2.55	0.34	
	Administrative staff	177	13.0	51.58	0.43		1.63	0.14		2.86	0.19	
	Others	81	5.9	51.53	0.65		1.80	0.23		2.68	0.28	
Total work years	0–5	497	36.5	52.21	0.26	0.032 *	1.45	0.08	**	2.30	0.11	0.006 *
	5–10	454	33.3	51.42	0.28		1.87	0.09		2.83	0.12	
	10–15	230	16.9	51.80	0.35		1.96	0.13		2.70	0.16	
	≥15	182	13.4	50.89	0.41		2.09	0.16		2.76	0.18	
Current work years	0–1	356	26.1	52.54	0.31	0.006 *	1.50	0.10	**	2.36	0.14	0.033 *
	1–3	427	31.3	51.70	0.28		1.63	0.08		2.52	0.12	
	3–10	501	36.8	51.16	0.25		2.01	0.09		2.85	0.11	
	≥10	79	5.8	51.33	0.56		1.97	0.22		2.66	0.29	
Smoking	Yes	295	21.6	51.47	0.35	0.591	2.00	0.12	0.026 *	2.35	0.13	0.056
	smoked but already quitted	118	8.7	51.43	0.56		1.90	0.18		2.47	0.22	
	No and never smoked	950	69.7	51.80	0.18		1.67	0.06		2.70	0.08	
Amounts of Cigarettes smoked	1–5	89	30.2	52.65	0.69	0.104	1.51	0.19	0.007 *	2.04	0.26	0.308
	6–10	85	28.8	51.22	0.48		1.88	0.21		2.29	0.23	
	11–20	92	31.2	51.07	0.63		2.37	0.22		2.67	0.23	
	>20	29	9.8	49.83	1.48		2.66	0.40		2.38	0.41	
Frequency of over drinking	everyday	6	0.6	53.83	4.42		3.33	0.99	**	3.50	1.65	0.584
	5–6 times/week	9	0.9	48.44	2.58	0.064	1.78	0.60		2.00	0.94	
	3–4 times/week	21	2.1	50.14	1.23		3.57	0.50		3.24	0.62	
	1–2 times/week	74	7.3	50.91	0.62		2.50	0.24		2.73	0.26	
	1–3 times per month	232	22.9	51.24	0.38		2.19	0.13		2.84	0.17	
	less than 1 time a month	669	66.2	51.98	0.21		1.61	0.07		2.61	0.10	
At least 30 min outdoor exercises	No	356	26.1	51.47	0.32	0.855	1.97	0.11	0.038 *	2.95	0.14	0.003 *
	1–3 times per month	537	39.4	51.81	0.23		1.71	0.08		2.53	0.11	
	1–3 times per week	226	16.6	51.92	0.38		1.47	0.12		2.14	0.15	
	4–6 times per week	129	9.5	51.46	0.54		1.85	0.17		2.78	0.23	
	Everyday	115	8.4	51.69	0.48		1.77	0.19		2.62	0.24	
At least 30 min high-intensity exercises	No	792	58.1	51.88	0.19	0.284	1.86	0.07	0.221	2.83	0.09	0.003 *
	1–3 times per month	392	28.8	51.21	0.30		1.66	0.10		2.30	0.13	
	1–3 times per week	128	9.4	52.17	0.46		1.50	0.16		2.17	0.20	
	4–6 times per week	38	2.8	51.63	1.24		1.66	0.33		2.55	0.47	
	Everyday	13	1.0	51.00	1.95		1.62	0.57		2.54	0.80	
At least 30 min medium-intensity exercises	No	770	56.5	51.74	0.20	0.927	1.92	0.07	0.004 *	2.83	0.09	0.002 *
	1–3 times per month	395	29.0	51.53	0.30		1.60	0.09		2.24	0.12	
	1–3 times per week	129	9.5	51.98	0.49		1.35	0.16		2.33	0.22	
	4–6 times per week	51	3.7	51.51	0.96		1.71	0.29		2.84	0.39	
	Everyday	18	1.3	52.17	0.98		1.33	0.44		2.39	0.68	
Work hours/week	<40	69	5.1	53.00	0.74	**	0.97	0.18	**	2.13	0.27	**
	40–49	734	53.9	52.64	0.20		1.16	0.06		2.36	0.09	
	50–59	254	18.6	51.02	0.35		2.09	0.12		2.69	0.16	
	≥60	306	22.5	49.72	0.32		3.11	0.11		3.22	0.15	
Extra work hours/day	0	318	23.3	54.03	0.32	**	0.66	0.07	**	2.00	0.13	**
	1–2	757	55.5	51.48	0.20		1.68	0.07		2.64	0.09	
	3–4	200	14.7	49.59	0.36		2.98	0.14		3.16	0.18	
	≥5	88	6.5	49.97	0.74		3.66	0.19		3.30	0.31	
Insomnia	No	868	63.7	51.91	0.19	0.075	1.46	0.06	**	2.26	0.08	**
	Yes	495	36.3	51.33	0.25		2.29	0.09		3.21	0.12	

Notes: Total/Score are calculated by mean value; *p* value marked with * shows it is less than 0.05 and *p* placed with ** means it is less than 0.001.

**Table 3 healthcare-11-02322-t003:** Correlation analysis relationship among occupational stress, cumulative fatigue, and WMSDs.

	Occupational Stress	Cumulative Fatigue	WMSDs
Occupational Stress	1.000	−0.329 *	−0.106 *
Cumulative fatigue	-	1.000	0.391 *
WMSDs	-	-	1.000

Note: * means the two indicators are significantly correlated (*p* < 0.05) by the two tailed *p* = 0.05.

**Table 4 healthcare-11-02322-t004:** Results of logistic regression for outcomes of each index.

	Occupational Stress			Cumulative Fatigue			WMSDs		
Variables	OR	95%CI	*p*	OR	95%CI	*p*	OR	95%CI	*p*
**Female Sex**	0.873	0.672–1.134	0.309	0.998	0.723–1.378	0.991	2.305	1.780–2.986	**
**Drinking Alcohol**							1.663	1.295–2.136	**
**Total Work Experience/years**									
5–10 (excluded 10)	1.037	0.771–1.395	0.812	1.139	0.753–1.722	0.539	1.086	0.775–1.522	0.630
10–15 (excluded 15)	1.130	0.782–1.634	0.515	1.319	0.695–2.504	0.397	1.207	0.716–2.034	0.480
≥15	1.359	0.9–2.052	0.145	1.307	0.583–2.932	0.516	2.032	1.060–3.896	0.033
**Current work experience/years**									
1–3 (excluded 3)	1.567	1.158–2.122	0.004	1.353	0.965–1.897	0.080	1.344	1.020–1.772	0.036
3–10 (excluded 10)	1.599	1.173–2.179	0.003	1.499	1.055–2.128	0.024	1.366	1.024–1.821	0.034
≥10	1.174	0.662–2.081	0.583	1.441	0.751–2.764	0.271	1.071	0.631–1.818	0.799
**Medium-intensive exercising**									
Occasionally, 1–3 times per month				0.727	0.542–0.975	0.033	0.789	0.593–1.048	0.102
Yes, 1–3 times per week				0.545	0.342–0.868	0.011	1.021	0.646–1.616	0.928
Usually, 4–6 times per week				0.958	0.483–1.900	0.902	0.969	0.476–1.974	0.932
Everyday				0.491	0.153–1.571	0.230	0.571	0.206–1.582	0.281
**high-intensive exercising**									
1–3 times per month							0.680	0.511–0.903	0.008
1–3 times per week							0.687	0.434–1.087	0.109
4–6 times per week							0.578	0.258–1.294	0.182
Everyday							1.084	0.345–3.411	0.890
**Weekly Work hours**									
40–49	1.441	0.836–2.483	0.188	1.228	0.669–2.255	0.507	1.189	0.740–1.911	0.475
50–59	1.707	0.945–3.083	0.076	2.221	1.152–4.282	0.017	1.143	0.675–1.934	0.619
≥60	2.767	1.512–5.065	0.001	3.668	1.871–7.191	**	1.571	0.917–2.690	0.100
**Extra Work hours/day**									
1–2	2.113	1.555–2.87	**	2.155	1.520–3.057	**	1.741	1.322–2.294	**
3–4	3.560	2.24–5.658	**	4.289	2.567–7.167	**	2.049	1.346–3.119	0.001
≥5	2.629	1.539–4.492	**	21.075	9.575–46.388	**	1.913	1.168–3.134	0.010
**Insomnia**				2.156	1.659–2.801	**	1.878	1.507–2.340	**

Notes: Being male, never drinking alcohol, being between 20 and 25 years old, being single, having a monthly income less than 465.63 USD, a total work experience of 0–5 years, current work experience of 0–1 years, 1–5 cigarettes smoked per day, no outdoor exercise, no high-intensity exercise, no medium-intensity exercise, weekly work hours less than 40, no extra work hours per day, no insomnia symptoms, no occupational stress, no cumulative fatigue are comparison characteristics. *p* placed with ** means it is less than 0.001.

**Table 5 healthcare-11-02322-t005:** Results of logistic regression for WMSDs, considering occupational stress and cumulative fatigue.

Variables	B	S.E.	Wald c	*p*	OR	95%CI
**Female Sex**	0.890	0.1341	44.015	**	2.435	1.872–3.166
**Drinking Alcohol**	0.467	0.1296	12.954	**	1.594	1.237–2.056
**Age**						
>25–30 (included 30)	−0.033	0.1827	0.033	0.857	0.968	0.676–1.384
>30–35	0.067	0.2592	0.067	0.796	1.069	0.644–1.777
>35	−0.455	0.3436	1.756	0.185	0.634	0.323–1.244
**Marital Status**						
Married and living together	0.224	0.1529	2.143	0.143	1.251	0.927–1.688
Married but separated	−0.190	0.2748	0.477	0.490	0.827	0.483–1.417
Divorced or widowed	−0.178	0.4540	0.153	0.695	0.837	0.344–2.038
**Monthly Income**						
3000–4999 yuan	−0.464	0.4056	1.311	0.252	0.629	0.284–1.392
5000–6999 yuan	−0.075	0.4042	0.034	0.853	0.928	0.42–2.049
7000–8999 yuan	−0.364	0.4126	0.777	0.378	0.695	0.31–1.561
9000–10,999 yuan	0.050	0.4242	0.014	0.905	1.052	0.458–2.416
>11,000 yuan	−0.075	0.4120	0.033	0.856	0.928	0.414–2.081
**Total Work Experience/years**						
5–10 (excluded 10)	0.060	0.1758	0.116	0.734	1.062	0.752–1.498
10–15 (excluded 15)	0.136	0.2718	0.251	0.616	1.146	0.673–1.952
≥15	0.689	0.3385	4.140	0.042	1.991	1.026–3.866
**Current work experience/years**						
1–3 (excluded 3)	0.238	0.1440	2.732	0.098	1.269	0.957–1.682
3–10 (excluded 10)	0.220	0.1504	2.145	0.143	1.246	0.928–1.674
≥10	−0.065	0.2741	0.056	0.813	0.937	0.548–1.604
**Outdoor exercising**						
Occasionally, 1–3 times per month	−0.069	0.1407	0.241	0.624	0.933	0.708–1.23
Yes, 1–3 times per week	−0.222	0.1773	1.562	0.211	0.801	0.566–1.134
Usually, 4–6 times per week	0.135	0.2111	0.409	0.523	1.145	0.757–1.731
Everyday	−0.011	0.2215	0.002	0.962	0.990	0.641–1.528
**high-intensive exercising**						
Occasionally, 1–3 times per month	−0.448	0.1485	9.107	0.003	0.639	0.478–0.855
Yes, 1–3 times per week	−0.418	0.2384	3.073	0.080	0.658	0.413–1.051
Usually, 4–6 times per week	−0.593	0.4156	2.033	0.154	0.553	0.245–1.249
Everyday	−0.013	0.5914	0.001	0.982	0.987	0.31–3.145
**Medium-intensive exercising**						
Occasionally, 1–3 times per month	−0.161	0.1482	1.183	0.277	0.851	0.637–1.138
Yes, 1–3 times per week	0.196	0.2394	0.668	0.414	1.216	0.761–1.944
Usually, 4–6 times per week	−0.009	0.3685	0.001	0.981	0.991	0.481–2.041
Everyday	−0.366	0.5265	0.484	0.486	0.693	0.247–1.945
**Weekly Work hours**						
40–49	0.139	0.2464	0.318	0.573	1.149	0.709–1.862
50–59	−0.054	0.2748	0.038	0.845	0.948	0.553–1.624
≥60	0.159	0.2828	0.314	0.575	1.172	0.673–2.04
**Extra Work hours/day**						
1–2	0.393	0.1442	7.434	0.006	1.482	1.117–1.966
3–4	0.385	0.2225	2.995	0.084	1.470	0.95–2.273
≥5	0.006	0.2628	**	0.983	1.006	0.601–1.684
Insomnia	0.482	0.1155	17.395	**	1.619	1.291–2.03
**Occupational Stress**	0.064	0.1139	0.315	0.575	1.066	0.853–1.333
**Cumulative fatigue**	1.165	0.1248	87.141	**	3.207	2.511–4.096

Notes: Being male, never drinking alcohol, being between 20 and 25 years old, being single, having a monthly income of less than 465.63 USD, a total work experience of 0–5 years, having been at the current work for 0 to 1 years, smoking 1–5 cigarettes per day, performing no outdoor exercise, no high-intensity exercise, no medium-intensity exercise, working less than 40 hours weekly, no extra work hours per day, no insomnia symptoms, no occupational stress, and no cumulative fatigue are comparison characteristics. *p* and Wald c placed with ** means it is less than 0.001.

## Data Availability

The data presented in this study are available on request from the corresponding author. The datasets generated and/or analyzed during the current study are not publicly available due to ethical agreements from participants.

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
