# Peer review of "The Prevalence and Correlated Factors of Occupational Stress, Cumulative Fatigue, and Musculoskeletal Disorders among Information Technology Workers: A Cross-Sectional Study in Chongqing, China"

_healthcare, 2023, doi:10.3390/healthcare11162322_

Round 1
Reviewer 1 Report
L171-173 „For the quantitative data that met the normal assumption“ Correct is “an assumption of normal distribution”
Please define RMB prior to using abrevations
Please avoid using terms like “was a risk factor” (L25-26, 45-46 and so on). This is a cross sectional study where only associations can be investigated. Please use terms like “was associated with increased risk”.
Table 4 , Variable SEX. Please change to ‘female sex’ or ‘male sex’, as it is not clear for what sex is OR corresponding
Please look at the numbers: Table 4 occurs twice (both tables are Table 4)
Please change p values 0.000 to <0.001
Author Response
Thanks for your careful review and helpful advice. Please see the attachment.

Reviewer 2 Report
Dear authors,
I have many concerns regarding the scientific methods and conclusions of your study. You must address at least the following topics to improve the scientific soundness of your work:
1. You use informal language all over your manuscript (e.g., didn't must be written as did not)
2. Introduction:
* Lines 48-58: you talk about your study, and then you dramatically change to another study. I do not find a connection between them. Has the other study encouraged your work? If so, the text must be reformulated.
* Overall, the text in this section must be improved. I suggest presenting the main points of the background/literature review. Only after that, describe the reasons that motivated/encouraged your work and the objectives for the study (or the topics addressed by the study).
* You present studies related to nurses and teachers. However, your study sample is very different. Can you provide studies regarding the occupational sectors included in the study sample?
* In this section, How do you support your background/literature search with only 10 references? Please, use more scientific references that address the topics covered in the introduction. Also, you should give more details about each topic. I have a few suggestions to add as references:
-> To stress:
- DOI: 10.1080/03055698.2020.1837080
- https://www.osha.gov/worker-fatigue
- https://doi.org/10.1007/s12310-022-09533-2
-> To WRMSD risk factors:
- https://doi.org/10.1007/978-3-030-89617-1_36
- https://osha.europa.eu/en/themes/musculoskeletal-disorders
3. Materials and methods:
* You present many results in the materials and methods section, such as the sample characterization.
* Section 2.3. should have a more specific name. I suggest "Assessment of occupational stress". The same suggestion to 2.4 and probably to 2.5.
* Did you develop the Chinese Musculoskeletal Disorders Questionnaire within this study? If yes, it must be carefully explained, detailed, and validated. If not, English must be improved to clarify.
* Please, provide scientific references to the statistical methods applied, such as Mann-Whitney and Kruskall-Wallis.
4. Discussion:
* How do you support that some participants have/had WRMSD? Did a physician diagnose them? Are MSD or WRMSD?
* How can you scientifically support that some participants have occupational stress based on the application of a self-assessment questionnaire? Do you have a physician's opinion?
* Discussion is very general. You presented a lot of tables with a lot of statistical parameters in the results section. However, most of them are not used, interpreted or even compared in the text.
5. Conclusion:
* conclusions are very few and general.
Please, read my previous comments.
Author Response
Thank you so much for your careful review and helpful advice! You raised most of the useful comments which make my manuscript look like a real “scientific article”. Please see the attachment.

Reviewer 3 Report
This is one those studies, in an increasing numbers, which research obvious facts and then try to wrap them as scientific results. I have made some comments on the attached file.

It is OK.
Author Response

(The authors gave the same response as above.)

Reviewer 4 Report
The authors aimed to examines the factors influencing occupational 15 stress, cumulative fatigue, and musculoskeletal disorders in the IT industry.
The scientific contents are not new.
The title has abbreviation 'IT' which is incorrect.
Few Keywords are incorrect. The authors need to check the MeSH database.
RMB should be converted to USD, especially when the manuscript is submitted to an international journal.
Regarding the subjects, it is better to state if living with family or colleagues or alone. Living alone and the frequency of communication with neighbours have also been shown to be associated with loneliness (Ministry of Health, Labour and Welfare of Japan: Decisions by the Headquarters for Novel Coronavirus Disease Control. Basic Policies for Novel Coronavirus Disease Control. https://www.mhlw.go.jp/content/10200000/000603610.pdf (2020).
How were the confounding factors adjusted?
Decision latitude with respect to work, psychological demands, support from supervisors, and support from co-workers are four key factors that determine job characteristics. These facts were left out in the discussion.
Previous studies showed that lower support, lower decision latitude, and higher psychological job demands can lead to stress and health issues. These facts need to be addressed.
According to the authors, what should be the ideal physical activity duration threshold for individuals?
Sleep quality, sleep latency, sleep duration, habitual sleep efficiency, sleep disturbance, use of sleeping pills, and daytime dysfunction may be necessary to arrive at any conclusion.
Can COVID-19 exposure influence the results of the study?
The tense is not uniform in the same paragraph of the text.
Example Refer to line 52- "The study reveals.....
Same paragraph, Line 55- "This study investigated.
Ideally, the tense should be same when used in the same paragraph.
Author Response

(The authors gave the same response as above.)

Round 2
Reviewer 2 Report
Dear authors,
I believe your manuscript has been clearly improved. Congratulations.
Author Response
Thanks again for your helpful advice and diligent check work. It's a pleasure to learn from your professional knowledge and dedicated attitudes. Hopefully we could have potential collaborations in the future!
Reviewer 4 Report
The authors could not address point No.5 and point.10 properly even in the revised version.
Acceptable.
Author Response
As for point 5, I already addressed that at the "Limitations" part in the previous revised manuscript. And for point 10, this time I also added that in the "Limitations" part. Thank you very much for your hard work. It's a pleasure for us to cooperate with you. Please see the attachment.
